# An elusive electron shuttle from a facultative anaerobe

Emily Mevers[1†], Lin Su[2,3†], Gleb Pishchany[1,4], Moshe Baruch[2], Jose Cornejo[2], Elissa Hobert[1], Eric Dimise[1], Caroline M Ajo-Franklin[2,5]*, Jon Clardy[1]*

[1]Department of Biological Chemistry and Molecular Pharmacology, Harvard Medical School, Boston, United States; [2]Molecular Foundry Division, Lawrence Berkeley National Laboratory, University of California, Berkeley, Berkeley, United States; [3]State Key Laboratory of Bioelectronics, School of Biological Science and Medical Engineering, Southeast University, Nanjing, China; [4]Department of Microbiology, Harvard Medical School, Boston, United States; [5]Molecular Biophysics and Integrated Bioimaging Division, Lawrence Berkeley National Laboratory, University of California, Berkeley, Berkeley, United States

**Abstract** Some anaerobic bacteria use insoluble minerals as terminal electron acceptors and discovering the ways in which electrons move through the membrane barrier to the exterior acceptor forms an active field of research with implications for both bacterial physiology and bioenergy. A previous study suggested that *Shewanella oneidensis* MR-1 utilizes a small, polar, redox active molecule that serves as an electron shuttle between the bacteria and insoluble acceptors, but the shuttle itself has never been identified. Through isolation and synthesis, we identify it as ACNQ (2-amino-3-carboxy-1,4-naphthoquinone), a soluble analog of menaquinone. ACNQ is derived from DHNA (1,4-dihydroxy-2-naphthoic acid) in a non-enzymatic process that frustrated genetic approaches to identify the shuttle. Both ACNQ and DHNA restore reduction of AQDS under anaerobic growth in menaquinone-deficient mutants. Bioelectrochemistry analyses reveal that ACNQ ($-0.32$ $V_{Ag/AgCl}$) contributes to the extracellular electron transfer (EET) as an electron shuttle, without altering menaquinone generation or EET related cytochrome c expression.
DOI: https://doi.org/10.7554/eLife.48054.001

*For correspondence:
cajo-franklin@lbl.gov (CMA-F);
jon_clardy@hms.harvard.edu (JC)

†These authors contributed equally to this work

## Introduction

Life is powered by redox reactions. It survives on the energy released as electrons move from the substrates that are oxidized to terminal electron acceptors (TEAs). The redox reactions that support life embrace a huge variety of substrates, intermediates, and acceptors as different environments demand different tactics. Anaerobic bacteria with an insoluble TEA face an especially demanding challenge as they must move electrons from inside their cells where the life-sustaining redox reactions occur, to outside the cell where the TEA resides. One solution to this transfer is an 'electron shuttle' – a redox active, diffusible molecule that shuttles electrons between a bacterial cell and a TEA (*Brutinel and Gralnick, 2012*; *Gralnick and Newman, 2007*).

*Shewanella oneidensis* MR-1 has been a model organism for studying extracellular electron shuttles since it was discovered by Myers and Nealson in 1988 for its ability to use an array of insoluble, extracellular TEAs (*Myers and Nealson, 1988*). In 2000 Newman and Kolter provided evidence for an extracellular electron shuttle in MR-1 by demonstrating that strains with a defect in menaquinone (MK) biosynthesis could not grow anaerobically with some TEAs, unless nearby wild type MR-1 provided a 'quinone-like' extracellular electron shuttle that functioned as a public good (*Newman and Kolter, 2000*). While their study suggested the existence of an electron shuttle, it did not identify it, and additional attempts at identification led to conflicting results. Subsequent studies by several

**eLife digest** In order to survive, we break down food through a series of chemical reactions that release energy to power our cells. In these metabolic reactions, small electrically charged particles called electrons are removed from the food molecule, and transferred, via a series of reactions, to a terminal electron acceptor.

For humans and many other organisms, oxygen is the terminal electron acceptor. Bacteria generate energy through a similar series of chemical reactions, but many species of bacteria live in environments where oxygen is absent. Some bacteria solve this problem by transferring the electrons released in their metabolic reactions to acceptor compounds in the external environment. These species must therefore employ a small molecule 'shuttle' to carry the electrons to the acceptor.

Previous work has shown the bacterial strain *Shewanella oneidensis* MR-1 releases a small molecule into its surrounding environment, which serves as its electron shuttle. Despite identifying a mutant strain of MR-1 that cannot produce this shuttle, researchers have been unable to determine the exact chemical identity of this critical molecule.

Now, Mevers, Su et al. have identified this elusive electron shuttle. This involved growing MR-1 and isolating the active molecule which restores the mutant bacteria's ability to shuttle electrons. Further experiments characterizing the structure of this compound using techniques involving analytical and synthetic organic chemistry revealed it be a small molecule known as ACNQ.

Mevers, Su et al. showed MR-1 produces this elusive electron shuttle by releasing a precursor structure into the environment where it spontaneously converts into ACNQ. As a result, there are no genes present in the genome of MR-1 or other bacterial strains that are required for the production of ACNQ. This genetic absence and low production levels of ACNQ has frustrated previous attempts to identify MR-1's electron shuttle.

Bacterial metabolism is studied for its applications in bioenergy (producing renewable energy using living organisms) and bioremediation (detoxification of substances using the reactions of bacterial metabolism). A better understanding of bacterial metabolism is thus essential for the continued development of these useful technologies.

DOI: https://doi.org/10.7554/eLife.48054.002

research groups have since demonstrated that excreted flavins by MR-1 can mediate the reduction of various insoluble TEAs (*Brutinel and Gralnick, 2012*; *Kotloski and Gralnick, 2013*; *Marsili et al., 2008*; *von Canstein et al., 2008*), such as Fe(III) oxide and electrodes. Conversely, a report by Myers and Myers in 2004 used thin-layer chromatography and UV absorption spectroscopy to conclude that the excreted 'quinone-like' metabolite was not an electron shuttle, but rather a biosynthetic intermediate of MK that is capable of bypassing the MK-deficient mutants' defect and thus generating small amounts of demethyl-MK (*Myers and Myers, 2004*). Collectively the previous attempts to identify the elusive electron shuttle demonstrated that it is small (<300 amu), very polar, potent, and excreted under both anaerobic and aerobic conditions by many different bacteria. We were motivated to reinvestigate the identity of this metabolite - shuttle or MK intermediate - as it was never characterized and its identity has reemerged as a research priority through its potential role in bioenergy development (*Shi et al., 2016*). In this report we provide evidence that ACNQ (2-amino-3-carboxy-1,4-naphthoquinone) acts as MR-1's elusive electron shuttle, describe the unusual biosynthetic pathway leading to ACNQ formation and demonstrate the ability of ACNQ to transfer electrons from MR-1 to carbon felt anodes.

## Results and discussion

We began by culturing MR-1 as previously described and assaying for shuttling activity using the original assay (*Newman and Kolter, 2000*) - the ability to recover respiration of an isogenic MK-deficient *menC* mutant strain (*menC*::Tn*10*) on a color-changing TEA, anthraquinone-2,6-disulfonate (AQDS), (*Figure 1A*). Rounds of bioassay-guided isolation (see Materials and methods – *Figure 1—figure supplement 1A,B*) produced highly active reduced complexity fractions and ultimately a single fraction containing the active molecule ($C_{11}H_8NO_4$, $[M + H]^+$ *m/z* 218.0445, calcd. 218.0448 Δ

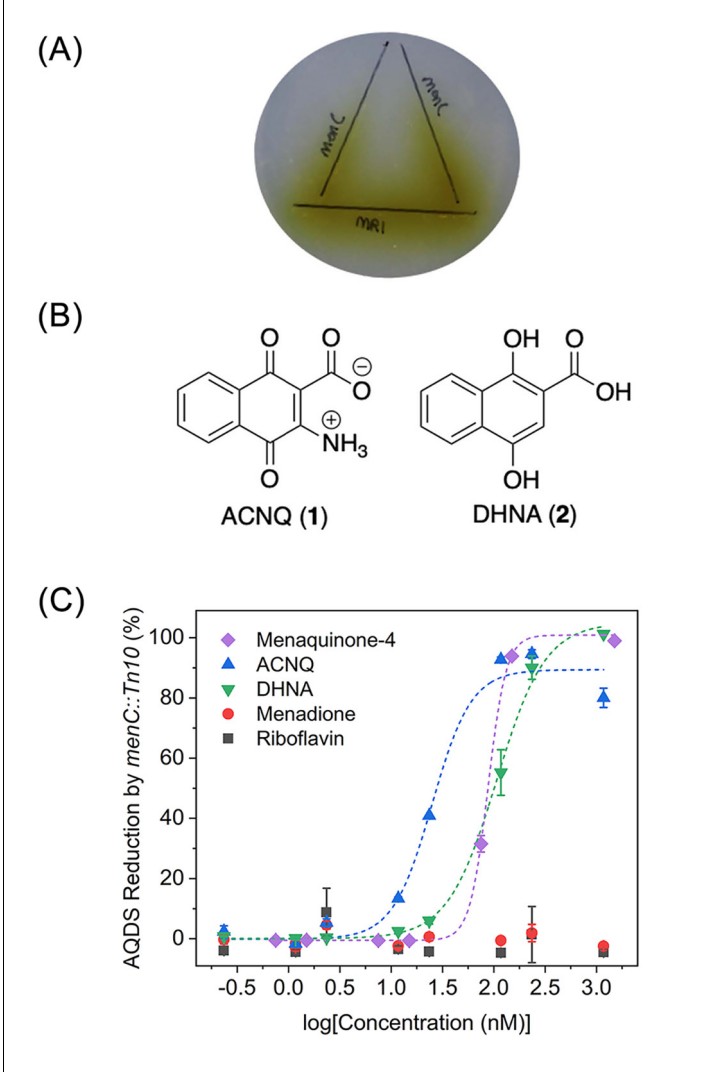

**Figure 1.** Functional analysis and structures of electron shuttles. (**A**) Wild type *S. oneidensis* MR-1 was inoculated on the horizontal line and *menC* mutant was inoculated on the vertical lines (x2). Reduction of AQDS (colorless) to AHQDS (yellow) is only observed by *menC* mutant where it was growing in close proximity to MR-1. (**B**) Structures of ACNQ and DHNA. (**C**) Dose-response of ACNQ, DHNA, menaquinone-4 (MK4), menadione, riboflavin to recover reduction of AQDS by *menC* mutant. Fitting with 'DoseResp' function by OriginPro 2019 (OriginLab Corporation).

DOI: https://doi.org/10.7554/eLife.48054.003

The following figure supplements are available for figure 1:

**Figure supplement 1.** Purification scheme of ACNQ from spent supernatant.

DOI: https://doi.org/10.7554/eLife.48054.004

**Figure supplement 2.** Comparison of positive mode fragmentation pattern of ACNQ isolated from MR-1 spent supernatant (bottom) and synthesized material (top).

DOI: https://doi.org/10.7554/eLife.48054.005

**Figure supplement 3.** Complete deuterium exchange on the LCMS, using mixtures of $D_2O$/acetonitrile as the mobile phase, revealed that the active component (ACNQ) has three exchangeable protons.

DOI: https://doi.org/10.7554/eLife.48054.006

**Figure supplement 4.** Comparison of negative mode fragmentation pattern of ACNQ isolated from MR-1 spent supernatant (top) and synthesized material (bottom).

DOI: https://doi.org/10.7554/eLife.48054.007

1.4 ppm). Detailed characterization was challenging as positive mode MS fragmentation did not match any metabolite in commercially available libraries (*Figure 1—figure supplement 2*), and our extremely mass-limited samples precluded NMR techniques. Additional MS-based experiments revealed that three of the seven protons were exchangeable (*Figure 1—figure supplement 3*), and negative mode MS data analyzed with the Metlin database (*Guijas et al., 2018*) identified a MK fragment (*Figure 1—figure supplement 4*). The candidate fitting the molecular composition, MK core, and other data was ACNQ (2-amino-3-carboxy-1,4-naphthoquinone, *Figure 1B*). An independent synthesis of ACNQ and LCMS comparison (*Figure 1—figure supplements 2* and *4*) confirmed that ACNQ was the unidentified metabolite in the wild type supernatant that recovered utilization of AQDS (−184 mV vs. SHE; *Aulenta et al., 2010*) by the *menC* mutant (*menC*::Tn*10*). In a functional assay, ACNQ potently recovers utilization of AQDS by *menC* mutant (*menC*::Tn*10*) with an $EC_{50}$ 25.0 ± 6.0 nM (*Figure 1C*).

Newman and Kolter reported that both *Vibrio cholerae* and other *Shewanella* species could recover respiration of the MK-negative mutant whereas *Pseudomonas fluorescens* could not (*Newman and Kolter, 2000*). In order to confirm that ACNQ is responsible for the recovery phenotype, we screened other bacterial strains for ACNQ production. ACNQ production was previously reported from *Propionibacterium freudenreichii* (*Kaneko, 1999*; *Kouya et al., 2007*; *Mori et al., 1997*), and we confirmed production in several Gram-negative bacteria – *V. cholerae*, *Escherichia coli*, and *Bacteroides fragilis* – and a Gram-positive bacterium – *Lactococcus lactis* (*Figure 2—figure supplements 1–2*). Quantitative analysis showed that MR-1 produces approximately 10–100 times more ACNQ than either *E. coli* or *V. cholerae* (*Figure 2A*). The ACNQ concentration in MR-1 supernatant under laboratory conditions is approximately 1.5 nM (*Figure 2—figure supplement 3*). Although this is lower than the calculated $EC_{50}$ value, simulation of ACNQ production and diffusion in an agarose gel indicates that local concentration of ACNQ when MR-1 forms a biofilm may reach 400 nM in two days (*Figure 2—figure supplement 4*).

Once the structure of the active metabolite was known, we addressed its biosynthesis by MR-1 and other bacteria. Based on ACNQ's structural similarity to MK and the genetic evidence that transposon mutations in *menC* or *menB* in MR-1 (*Myers and Myers, 2004*; *Newman and Kolter, 2000*) and *E. coli* (*Figure 2—figure supplement 5*)] led to loss of production of ACNQ, it seemed likely that ACNQ was produced largely by the known MK biosynthetic pathway (*Figure 2B*). As mutations upstream of the synthesis of DHNA (1,4-dihydroxy-2-naphthoic acid) abolish ACNQ production, we hypothesized that DHNA is used as the substrate for ACNQ production (*Figure 2C*). *menA* mutants derived from MR-1 and *E. coli* K12 (*menA*::Himar and *menA789*::*kan*, respectively) produced 2.5 times higher amounts of ACNQ than their wild type parent counterparts (*Figure 2A* and *Figure 2—figure supplement 5*). This increase is consistent with the function of MenA in converting DHNA to demethyl-MK and thus reducing the levels of DHNA available for conversion to ACNQ. Additional support came from the observation that while the genes for the production of MK are not clustered, the genes for the production of DHNA are in one small region of the circular chromosome, while *menA*, which encodes a protein carrying out the step directly following DHNA production in the canonical pathway, is on the opposite side of the chromosome (*Figure 2B*).

To further characterize the conversion of DHNA to ACNQ we searched for the transaminase responsible for this reaction. The most likely candidate seemed to be GlmS, a glutamine:fructose-6-phosphate transaminase, which is expressed by a gene located downstream of *menB*. However, knocking out *glmS* did not result in loss of production of ACNQ (*Figure 2—figure supplement 6*). As there were 18 other transaminases in the NCBI-annotated MR-1 genome, we adopted a systematic approach using *E. coli* K12 rather than a piecemeal approach using MR-1 to identify the responsible transaminase. Strains containing disruptions in all 22 annotated genes encoding aminotransferases/transaminases found in *E. coli* K12 genome were acquired from The Coli Genetic Stock Center (CGSC), (*Supplementary file 1B*). All of the mutant strains were able to make ACNQ (*Figure 2D*). The failure to identify an *E. coli* aminotransferase/transaminase mutant unable to make ACNQ coupled with the failure of several prior genetic screens (transposon libraries) (*Myers and Myers, 2004*; *Newman and Kolter, 2000*) in MR-1 background to identify respiration defects in any genes other than MK biosynthetic genes suggested that DHNA might be converted to ACNQ by a non-enzymatic pathway.

To investigate whether ACNQ forms spontaneously, dilute DHNA was incubated in a variety of aqueous solutions, including the minimal media used in this work to culture MR-1, minimal media

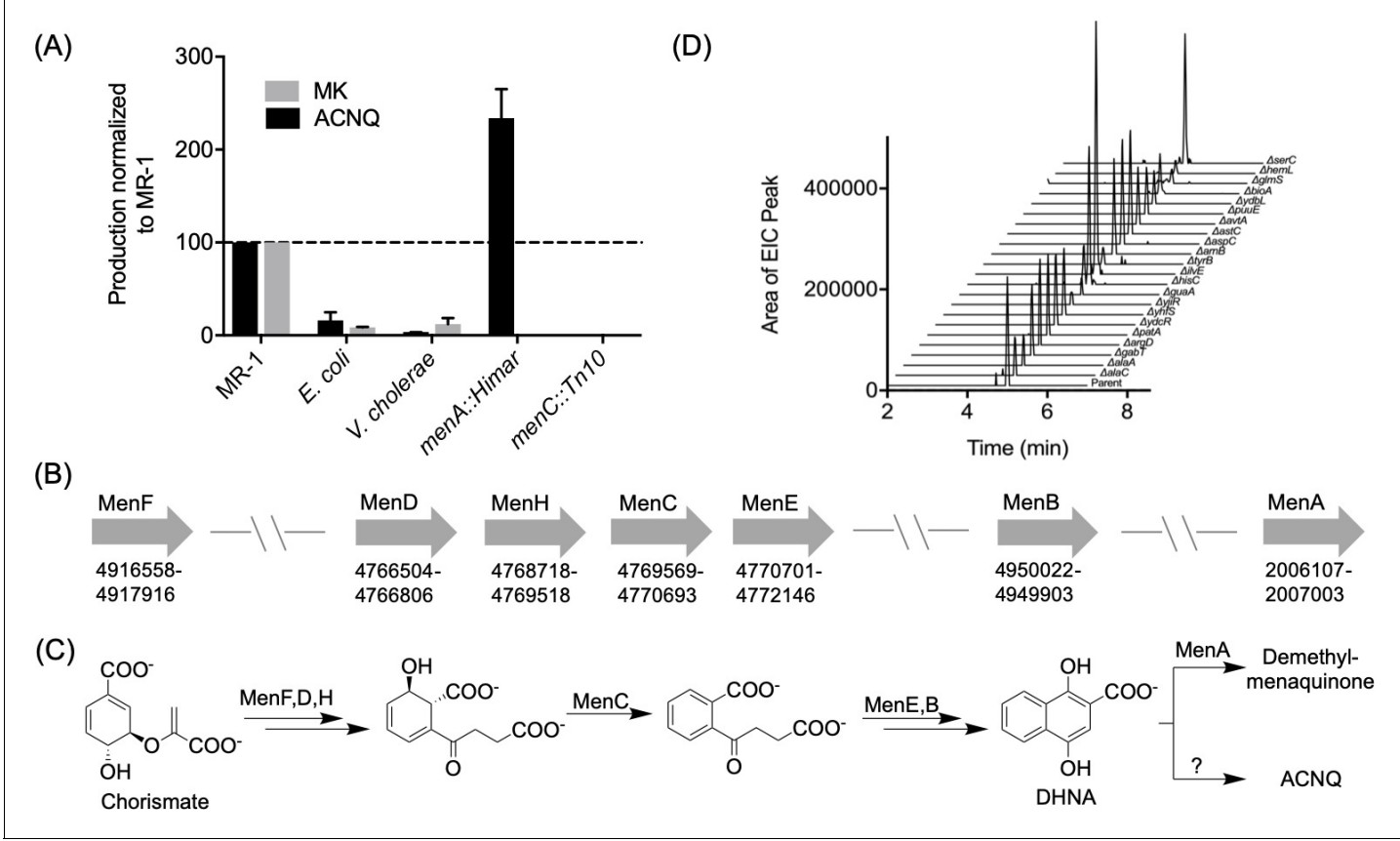

**Figure 2.** Production of and biosynthetic pathway for ACNQ. (A) Relative production levels of ACNQ and menaquinones by *S. oneidensis* MR-1 (MR1), *E. coli*, *V. cholerae*, *S. oneidensis menA*, and *menC* mutants. The ability of *E. coli* to produce ACNQ is in contrast to the original report by Newman and Kolter, however *E. coli* produces significantly lower quantities of ACNQ than *S. oneidensis*, which is likely below the detection limit of the technique used in the previous report. (B) Genes responsible for the biosynthesis of menaquinone and their location on the chromosome. (C) Biosynthesis scheme of dimethyl-menaquinone and ACNQ. (D) Production of ACNQ by all *E. coli* aminotransferase/transaminases knockout mutants.

DOI: https://doi.org/10.7554/eLife.48054.008

The following figure supplements are available for figure 2:

**Figure supplement 1.** Production of ACNQ by *V. cholerae* V52, *Lactococcus lactis lactis* (ATCC 25285), *E. coli* K12, and *Bacteroides fragilis* (DSM 20481).

DOI: https://doi.org/10.7554/eLife.48054.009

**Figure supplement 2.** Various bacterial strains that can recover reduction of AQDS by *S. oneidensis menC::Tn10*.

DOI: https://doi.org/10.7554/eLife.48054.010

**Figure supplement 3.** Absolute quantification of ACNQ in MR-1 culture using $^{15}$N-ACNQ as an internal standard to determine recovery yields.

DOI: https://doi.org/10.7554/eLife.48054.011

**Figure supplement 4.** Simulation of ACNQ production and diffusion in an agarose gel.

DOI: https://doi.org/10.7554/eLife.48054.012

**Figure supplement 5.** Production of ACNQ by various *E. coli* K12 *men* mutants.

DOI: https://doi.org/10.7554/eLife.48054.013

**Figure supplement 6.** Production of ACNQ by *S. oneidensis ΔglmS*.

DOI: https://doi.org/10.7554/eLife.48054.014

**Figure supplement 7.** DHNA conversion to ACNQ in the same minimal media that was used to culture MR-1.

DOI: https://doi.org/10.7554/eLife.48054.015

**Figure supplement 8.** DHNA conversion to ACNQ in ammonium phosphate, pH 8.

DOI: https://doi.org/10.7554/eLife.48054.016

**Figure supplement 9.** DHNA conversion to ACNQ in minimal media with amino acids as the only nitrogen source.

DOI: https://doi.org/10.7554/eLife.48054.017

**Figure supplement 10.** Production of ACNQ when grown with (A) $NH_4SO_4$, (B) $NH_4SO_4$ and amino acid mixture and (C) amino acid mixture as the sole nitrogen source.

*Figure 2 continued on next page*

*Figure 2 continued*

DOI: https://doi.org/10.7554/eLife.48054.018

**Figure supplement 11.** Absolution quantification of menaquinone (MK) in MR-1, *V. cholerae* V52, and *E. coli* K12 cultures.
DOI: https://doi.org/10.7554/eLife.48054.019

**Figure supplement 12.** Proposed mechanism for the non-enzymatic conversion of DHNA to ACNQ.
DOI: https://doi.org/10.7554/eLife.48054.020

**Figure supplement 13.** Menaquinone analysis of *S. oneidensis menC::Tn10* (menC) and *S. oneidensis* MR-1 (MR-1) cell pellets when grown with 0.05, 0.5, and 1 µM of ACNQ supplemented to the media under anaerobic conditions with fumarate as the sole TEA.
DOI: https://doi.org/10.7554/eLife.48054.021

**Figure supplement 14.** Menaquinone analysis of *S. oneidensis menC::Tn10* (menC) and *S. oneidensis* MR-1 (MR-1) cell pellets when grown with 0.05 µM of ACNQ supplemented to the media under anaerobic conditions with TMAO as the sole TEA.
DOI: https://doi.org/10.7554/eLife.48054.022

**Figure supplement 15.** Menaquinone analysis of *S. oneidensis menC::Tn10* (menC) and *S. oneidensis* MR-1 (MR-1) cell pellets when grown with 0.05, 0.5, and 1 µM of ACNQ supplemented to the media under aerobic conditions.
DOI: https://doi.org/10.7554/eLife.48054.023

with a mixture of amino acids as the sole nitrogen source, and an ammonium phosphate buffered solution. All conditions led to the conversion of DHNA to ACNQ after 24 hr with no remaining DHNA detected (*Figure 2—figure supplements 7–9*). DHNA is quite labile and appeared to react with other media components as was evident by the complex LCMS chromatogram of the reaction mixtures. Furthermore, MR-1 produced ACNQ when grown on a variety of nitrogen sources, i.e. ammonium sulfate or different amino acid mixtures (*Figure 2—figure supplement 10*). Quantification of both the MK pool in the cell pellet and ACNQ in the supernatant, 350 nM and 1.5 nM of MR-1 culture, respectively, indicated that only 2% of the DHNA pool would need to be diverted to ACNQ production if ACNQ were to be generated entirely non-enzymatically (*Figure 2—figure supplement 11*). A likely non-enzymatic mechanism would commence with the oxidation of DHNA to the quinone then conjugate addition of ammonium at the C3 position followed by enol formation and a subsequent oxidation of the hydroquinone (*Figure 2—figure supplement 12*).

ACNQ, DHNA, menadione, MK-4, and riboflavin were each evaluated for their ability to recover *menC* mutant (*menC*::Tn*10*) reduction of AQDS to AHQDS under anaerobic conditions when it is the sole TEA. These metabolites where chosen because they represent either products of the disrupted pathway in the *menC* mutant strain (MK-4 and DHNA), were previously implicated in electron shuttling (riboflavin) or to define structure-activity relationships (menadione). ACNQ, DHNA, and MK-4 exhibited activity with $EC_{50}$ values of $25.0 \pm 6.0$, $106.2 \pm 11.2$, and $87.8 \pm 4.7$ nM, respectively (*Figure 1C*), while neither menadione nor riboflavin allowed for AQDS reduction by *menC* mutant strain (concentrations up to 50 µM). It should be noted that while DHNA can recover respiration nearly as efficiently as ACNQ, DHNA decomposes relatively quickly in media and was not detected in the spent supernatant of MR-1 cultures (data not shown). Although ACNQ is structurally similar to the MK precursor DHNA, addition of ACNQ does not replenish the MK pool in *menC* mutant (*menC*::Tn*10*) as we demonstrated by HR-LCMS analysis on the lipid contents of both MK-negative and MR-1 cultures grown in the presence of various concentrations of ACNQ. This analysis revealed no evidence that ACNQ was being converted to MK (*Figure 2—figure supplements 13–15*), demethyl-MK, or an amino-modified MK analog. The instrument limit of detection for MK and demethyl-MK under the tested conditions is approximately 60 fmol, allowing for the detection of the presence of approximately 20 pmol of MK in a 50 mL culture. The inability of ACNQ to be converted to MK or related analog contrasts with previous findings by Myers and Myers (*Myers and Myers, 2004*), however it is important to note that the previous work used a complex mixture of metabolites that is present in spent supernatant and used thin-layer chromatography, a much less informative analytical method.

To better understand the ability of ACNQ to act as an electron shuttle, we followed the work of Marsili et al. (*Marsili et al., 2008*) and performed detailed bioelectrochemical experiments in which we varied the availability of ACNQ. We grew a biofilm of MR-1 using lactate as the sole electron donor, M9 medium as the electrolyte with 1 µM ACNQ as the possible electron shuttle, and a carbon felt electrode biased at $-0.1$ $V_{Ag/AgCl}$ as the sole electron acceptor. This potential was chosen in order to restrict the redox processes close to the redox potential of ACNQ ($-0.32$ $V_{Ag/AgCl}$,

*Figure 3—figure supplement 1A*). The established biofilm produced a significant oxidation current,~22 µA·cm$^{-2}$, after 25 hr (*Figure 3A*). When the medium was removed and replaced with fresh anaerobic medium, the oxidation current dropped more than 90% [*Figure 3A*, curve (1)], showing that the major carrier of current was soluble. 1 µM ACNQ was then added into the reactor, resulted in an immediate increase in current [*Figure 3A*, curve (2)], which stabilized in 6 hr to 45% of the original biofilm current, indicating that ACNQ is responsible for the current generation. The loss of planktonic cells and secreted flavins in the original media when changing the media might be the reason that current is not fully restored. Cyclic voltammetry (CV) analysis [*Figure 3B*, curve (1) and (2)] reveals a major catalytic wave initiated at around −0.34 V$_{Ag/AgCl}$, which matches the oxidation peak potential of ACNQ. No peaks associated with flavins were observed under these conditions. Also, the peak width at half height (W$_{1/2}$) obtained by differential pulse voltammetry (DPV) is 66 mV (*Figure 3—figure supplement 1A*), indicating that more than one electron is transferred during the redox reaction as would be expected for the oxidation of a hydroquinone to quinone.

To determine the role of ACNQ more accurately, we performed a second replacement of the media with fresh media lacking ACNQ. The current decreased by 95% [*Figure 3A*, curve (3)]. Next, we re-introduced the original ACNQ-containing media after removing the planktonic cells through two rounds of centrifugation and filtration through a 0.22 µm membrane. The current immediately increased and recovered to 85% of the current before second media replacement [*Figure 3A*, curve (4)]. CV analysis [*Figure 3B*, curve (3) and (4)] shows the major catalytic wave starts at the same ACNQ-related potential. *Shewanella oneidensis* mutants that are unable to secrete flavins (Δ*bfe*) (*Kotloski and Gralnick, 2013*) or lack outer membrane cytochromes (Δ*mtr*) (*Coursolle and Gralnick, 2012*) show similar chronoamperometry (*Figure 3—figure supplement 2A,B*) and CV (*Figure 3—figure supplement 2C,D*) results, indicating EET by ACNQ does not require outer membrane cytochromes or flavins. Further CV analysis (*Figure 3—figure supplement 1B*) indicates that ACNQ is also acting as an absorbed redox species, much like what was observed with flavins (*Marsili et al., 2008*). These data indicates that ACNQ can act as the major electron carrier to a carbon-based electrode when biased at −0.1 V$_{Ag/AgCl}$.

A series of chronoamperometry experiments was performed to investigate how ACNQ influences extracellular electron transfer. In the absence of bacterial cells, ACNQ does not generate current to a biased electrode (*Figure 3—figure supplement 1C*). Likewise, in the absence of ACNQ, both wild type MR-1 and *menC::Tn10* produce very low current density (*Figure 3C*). We suggest that this occurs because the −0.1 V$_{Ag/AgCl}$ potential is too negative for electron transfer through outer membrane cytochromes, and because the bacteria were washed before injection so neither flavins nor ACNQ is present at sufficient concentration to mediate electron transfer. Addition of exogenous ACNQ to wild type MR-1 and *menC* mutant (*menC*::Tn*10*) immediately increases current 33- and 36-fold, respectively (*Figure 3C*). In accordance with our previous results, HR-LCMS analysis of the relative MK concentration revealed no differences in MK or amino-modified MK analogues' levels between ACNQ and DMSO-treated (*Figure 3—figure supplement 3*). Furthermore, ACNQ does not affect the cytochrome *c* expression in *menC* mutant (*Figure 3—figure supplement 1D*), which excludes the possibility that the current increase was due to any change in the cyt c. These observations strongly suggest that ACNQ can act to transfer electrons to extracellular TEAs in *S. oneidensis*.

Although defining ACNQ's precise role(s) in extracellular electron transfer will require future studies, in this work we demonstrated that it is the unidentified quinone-like metabolite whose existence was established almost 20 years ago. We confirmed that ACNQ is excreted from *S. oneidensis* MR-1 (and other bacteria), is responsible for AQDS reduction, and is capable of increasing current to an electrode via a mechanism that does not require menaquinone or outer membrane cytochrome *c*. The identification of ACNQ could prove useful in future bioenergy applications and should promote an examination of the roles flavins and ACNQ have as extracellular electron shuttles under different conditions.

## Materials and methods

### General experimental Procedures

UV spectra were measured on an Amersham Biosciences Ultrospec 5300-pro UV/Visible spectrophotometer and IR spectra were measured on a Bruker Alpha-P spectrophotometer. NMR spectra were

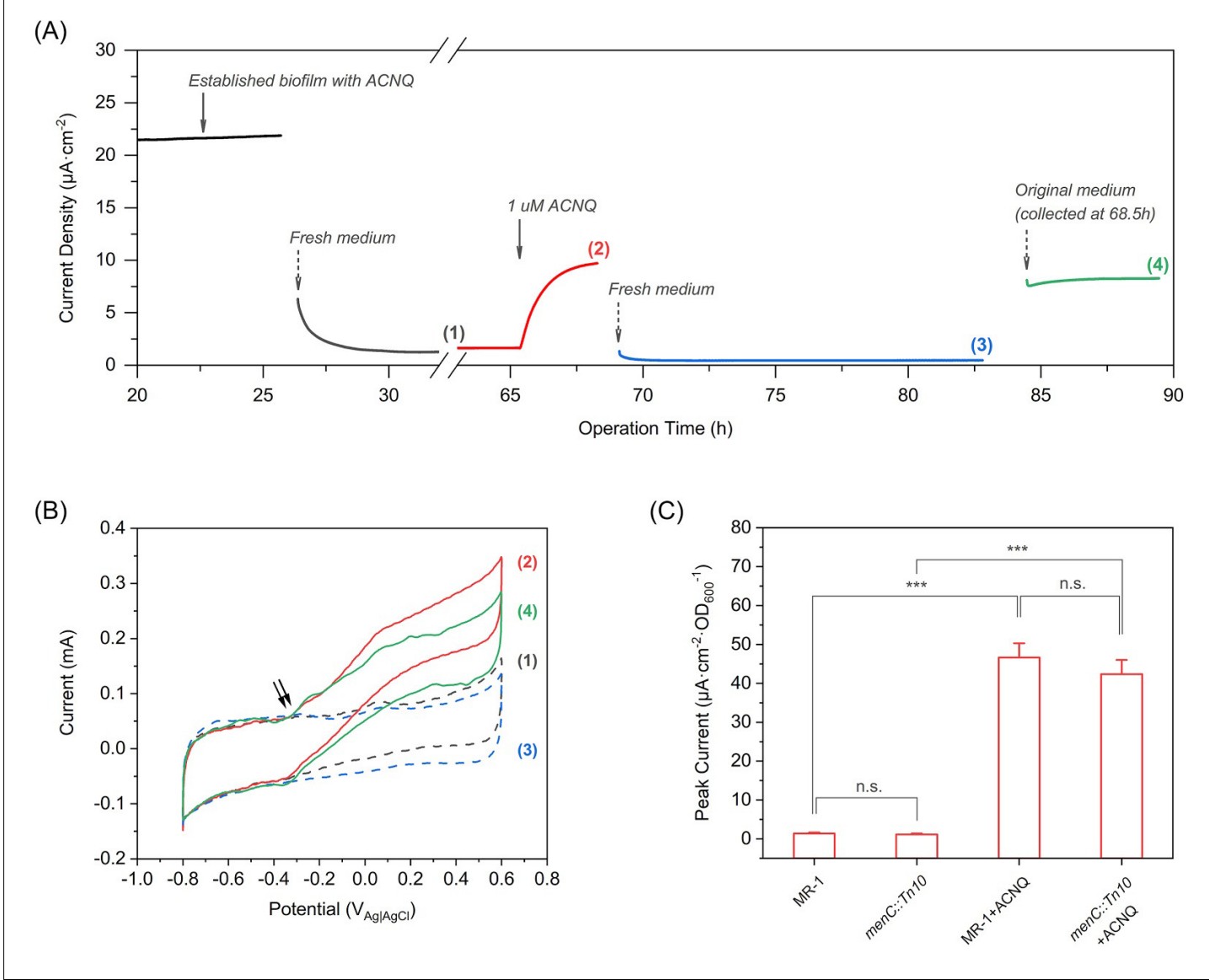

**Figure 3.** ACNQ acts as an extracellular electron transfer mediator for MR-1 in a bioelectrochemical system. (**A**) The current production of an established MR-1 biofilm over time in the presence of different media. Oxidation current drops after the introduction of fresh medium (gray trace), but recovers immediately after the addition of 1 μM ACNQ (red trace). The second replacement of fresh media (blue trace) again causes the current to drop, and the introduction of original ACNQ-containing media after filtration (green trace) leads to high oxidation current. Numbered gaps indicate CV analysis. (**B**) Turnover CV scan of the MR-1 biofilm with different media where the color and number indicate when the analysis was performed in(A). The double arrow highlights the initiation of the catalytic wave. (**C**) Peak current (steady state current, collected after at least 12 hr operation) from chronoamperometry experiments (electrode bias at $-0.1$ $V_{Ag/AgCl}$) with wild type *S. oneidensis* (MR-1) and *menC* mutant (*menC::Tn10*), with or without 1 μM ACNQ. ***$p<0.001$, ANOVA with Tukey Test by OriginPro 2019 (OriginLab Corporation), n = 3.

DOI: https://doi.org/10.7554/eLife.48054.024

The following source data and figure supplements are available for figure 3:

**Source data 1.** Source Data for *Figure 3C*.
DOI: https://doi.org/10.7554/eLife.48054.028

**Figure supplement 1.** ACNQ acts as an electron shuttle and does not affect cytochrome *c* protein expression.

DOI: https://doi.org/10.7554/eLife.48054.025

**Figure supplement 2.** ACNQ acts as an extracellular electron transfer mediator for *S. oneidensis* Δ*bfe* and Δ*mtr*.

DOI: https://doi.org/10.7554/eLife.48054.026

**Figure supplement 3.** Menaquinone analysis of *S. oneidensis menC::Tn10* (menC) and *S. oneidensis* MR-1 (MR-1) cell pellets when grown with 0.05, 0.5, and 1 μM of ACNQ supplemented to the media under aerobic conditions.

*Figure 3 continued on next page*

*Figure 3 continued*

DOI: https://doi.org/10.7554/eLife.48054.027

recorded with methanol or DMSO as an internal standard ($\delta_H$ 2.50 and $\delta_C$ 39.5) on a Bruker Avance 1 500 MHz spectrometer equipped with a TXO Helium CryoProbe (500 and 125 MHz for $^1$H and $^{13}$C NMR, respectively). LR-LCMS data were obtained using an Agilent 1200 series HPLC system equipped with a photo-diode array detector and a 6130-quadrupole mass spectrometer. HRESIMS was carried out using an Agilent 6530 LC-q-TOF Mass Spectrometer equipped with an uHPLC system. HPLC purifications were carried out using Agilent 1100 or 1200 series HPLC systems (Agilent Technologies) equipped with a photo-diode array detector. All solvents were of HPLC quality.

## Strain and general culture conditions

*Shewanella oneidensis* MR-1, an isogenic *menC* mutant (*menC*::Tn10), and *Vibrio cholerae* V52 were obtained from the Kolter lab at Harvard Medical School (*Newman and Kolter, 2000*). *S. oneidensis* strain with a *menA*::Himar mutation derived from MR-1 was kindly provided to us from the Barstow group at Princeton University (*Baym et al., 2016*), *S. oneidensis bfe* and *mtr* mutant strains (Δ*bfe* and Δ*mtr*) were provided by the Gralnick group at the University of Minnesota (*Coursolle and Gralnick, 2012*; *Kotloski and Gralnick, 2013*). *glmS* knockout strain in MR-1 background was constructed as part of this study (see below). Keio *E. coli* strains were purchased from the Coli Genetic Stock Center (CGSC) at Yale University (see *Supplementary file 1B*). *Bacteroides fragilis* ATCC 25285 and *Lactococcus lactis lactis* (DSM 20481) were purchased from commercial sources. All overnight cultures were grown in LB media (DB Difco) at 30°C, 200 rpm and all experimental cultures were grown at 30°C for 72 hr in the following lactate minimal media unless specifically stated otherwise: 1.19 g/L $(NH_4)_2SO_4$, 0.99 g/L $K_2HPO_4$, 0.45 g/L $KH_2PO_4$, 0.168 g/L $NaHCO_3$, 54 mg/L $CaCl_2$, 4.8 g/L disodium fumarate, 0.1 g/L vitamin-free casamino acids, 0.247 g/L $MgSO_4 \cdot 7H_2O$, 1.6 g/L sodium lactate, 100 mg/L l-Arg-HCl, 100 mg/L l-Glu, 100 mg/L l-Ser, and 1 mL/L of the following trace elements solution with the pH adjusted to 7.4. Trace element solution: 25.0 g/L $Na_2$-EDTA, 1.5 g/L $FeSO_4 \cdot 7H_2O$, 0.29 g/L $ZnSO_4 \cdot 7H_2O$, 0.22 g/L $MnSO_4 \cdot H_2O$, 3.50 g/L $H_3BO_3$, 1.41 g/L $CoSO_4 \cdot 7H_2O$, 0.05 g/L $CuSO_4 \cdot 5H_2O$, 1.97 g/L $Ni(NH_4)_2SO_4 \cdot 6H_2O$, 0.94 g/L $Na_2MoO_4 \cdot 2H_2O$, 0.28 g/L $Na_2SeO_4$, and 0.58 g/L NaCl.

## Isolation, Purification, and Identification of ACNQ

For purification of the active metabolite, *S. oneidensis* MR-1 was grown in batches of 16 × 4 L Erlenmeyer flasks containing 1 L of lactate minimal liquid media. After 40 min sterilization cycle each Erlenmeyer flask was cooled to room temperature (rt) overnight and then inoculated with 1 mL of a 5 mL overnight LB culture of MR-1 (grown at 30°C with shaking at 200 rpm). The Erlenmeyer flasks were then incubated at 30°C for 72 hr without shaking. Conditioned supernatant was separated from cells by centrifugation (14,000 rcf, 30 min), filter-sterilized (0.2 μm, Thermo Scientific, cat #: 595–4520), and acidified to a pH of 6.5 with 12.1 M HCl (*aq*). This material was then passed through a glass column containing 1 g of Strata-X-A resin (Phenomenex, 33 μm, 85 Å) that was prepped by washing with 15 mL of 100% MeOH and subsequently 15 mL of 100% $H_2O$. Upon passing all of the conditioned supernatant through the column, the resin was washed with 12 mL of 25 mM ammonium acetate (*aq*) pH 6.5% and 100% methanol. The crude material was eluted from the column with 15 mL of 5% formic acid (FA) in acetonitrile. The solvent then was immediately removed under vacuum to yield the crude extract.

Activity-guided isolation of the crude extract by reverse-phase (RP) high performance liquid chromatography (HPLC) yielded a potently active reduced-complex fraction. This active fraction eluted near the 33rd min using Phenomenex 4 μm Hydro semi-preparative column (250 × 10 mm), under the following conditions: hold 20% ACN +0.1% FA/$H_2O$ + 0.1% FA for 5 min then gradient to 50% ACN +0.1% FA/ $H_2O$+0.1% FA over 35 min with a flow rate of 3 mL/min. This reduced complex fraction was then analyzed on HRLCMS using a Phenomenex Kinetex 2.6 μm EVO C18 100 Å (100 × 2.1 mm) under the following LC method: hold 10% ACN +0.1% FA/$H_2O$ + 0.1% FA for 1 min followed by a gradient to 100% ACN +0.1% FA over 9 min then hold 100% ACN +0.1% FA for 3.5 min with a constant flow rate of 0.3 mL/min. For MS and MS/MS measurements the electrospray ionization (ESI)

parameters were set to 10 L/min sheath gas flow, 3.0 L/min auxiliary gas flow, and 325°C gas temperature. The spray voltage was set to 3.5 kV, the inlet capillary was set to 300°C and a 65 V S-lens radio frequency (RF) level was applied. MS/MS spectra were recorded in Auto MS/MS mode using a preferred ion list only. Scans were acquired at a rate of 13,577 transients/spectrum with a mass range of 50–1,700 $m/z$. The collision energy was set to 10, 20, and 30 eV for all ions.

### Deuterium exchange analysis

Analysis of the active reduced-complex HPLC fraction by the same HR-LCMS described above, except exchanging $D_2O$ (Sigma Aldrich) for $H_2O$ in the mobile phase, allowing for the determination of the exact number of exchangeable hydrogens the active metabolite contained. An aliquot (5 µL) of the reduced-complex HPLC fraction was analyzed using this modified method and a 3 Da mass shift was observed for both $[M + D]^+$ and $[M + Na]^+$ ion adducts. This indicated that the active metabolite possessed three exchangeable hydrogens.

### Synthesis and Large-Scale purification of ACNQ

Ammonium chloride (20 g, 0.374 mol) was added to a glass jar containing 2 L of deionized (DI) water and a magnetic stir bar. Once the solid dissolved, the pH of the mixture was adjusted to 9.0 using ~10 mL of ammonium hydroxide solution. Then 1,4-dihydroxy-2-naphthoic acid (DHNA - 1.0 g, 4.5 mmol) was added to the mixture. The reaction mixture was stirred for 24 hr at rt, at which point the mixture was acidified to pH five and then extracted with 2 L ethyl acetate (EtOAc) x 2. The organic layer was concentrated under vacuum. The crude material was then resuspended in 10 mL 50% ACN/$H_2O$, syringe filtered (Corning Inc, 0.2 µm), and purified using prep-HPLC. Pure ACNQ eluted between 33–35 min using Phenomenex Luna 5 µm C8(2) 100 Å preparative column (250 × 21.2 mm), with the following method: holding 20% ACN +0.1% FA/$H_2O$ + 0.1% FA for 5 min then gradient to 55% ACN +0.1% FA/$H_2O$ + 0.1% FA over 35 min at 10 mL/min flow rate. The overall reaction yield, through purification, is approximately 1%. Synthetic ACNQ was compared to the isolated material by back-to-back injection on the HRLCMS using a Phenomenex Kinetex 2.6 µm EVO C18 100 Å (100 × 2.1 mm) under the following LC method: holding 2.0% ACN +0.1% FA/$H_2O$ + 0.1% FA for 1 min then gradient to 20% ACN +0.1% FA/$H_2O$ + 0.1% FA over 39 min at 0.3 mL/min.

### Abiotic production of ACNQ

An aliquot of DHNA (15 µL of 10 mM solution in MeOH) was added to 10 mL of either lactate minimal media described above, lactate minimal media with an amino acid mixture replacing the ammonium sulfate [amino acid mixture (10x mixture): L-arginine: 1.26 g/L, L-cystine: 0.24 g/L, L-histidine HCl: 0.42 g/L, L-isoleucine: 0.52 g/L, L-leucine: 0.52 g/L, L-lysine HCl: 0.73 g/L, L-methionine: 0.15 g/L, L-phenylalanine: 0.33 g/L, L-threonine: 0.48 g/L, L L-tryptophan: 0.1 g/L, L-tyrosine: 0.36 g/L, L-valine: 0.467 g/L], and ammonium phosphate buffered aqueous solution, pH 8. The mixtures of samples were kept under aerobic conditions at rt, with 5 µL of each mixture analyzed on the LC-q-TOFMS for the production of ACNQ and consumption of DHNA at specific time points.

### 2-Amino-3-carboxy-naphthoquinone (ACNQ - 1)

Amorphous yellow solid; UV (MeOH) $\lambda_{max}$ (log ε) 207 (3.82), 232 (4.04), 266 (4.05), 334 (3.30), and 416 (3.25) nm; IR (KBr) 3294, 3173, 1684, 1593, 1542, 1479, 1462, 1395, 1367, 1319, 1286, 1232, 907.3, 775, 725, 695, 658, and 583 cm$^{-1}$; $^1$H NMR (500 MHz, $d_6$-DMSO) 14.5 (s), 10.0 (bs), 9.6 (bs), 8.12 (d; 7.8 Hz), 8.06 (d; 7.7 Hz), 7.93 (t; 7.5 Hz), and 7.83 (t; 7.83 Hz) ppm; $^{13}$C NMR (75 MHz, $d_6$-DMSO) 184.1, 178.9, 169.3, 155.8, 135.8, 133.8, 132.3, 130.3, 126.7, 126.5, and 96.9 ppm; HRESIMS $[M + H]^+$ $m/z$ 218.0456 (calcd. for $C_{11}H_8O_4N$ 218.0453, Δ 1.2 ppm).

### Cellular Respiration Assay with Anthraquinone-2,6-disulfonate (AQDS)

To follow the activity through purification, we used a modified method of the AQDS assay previously described (*Newman and Kolter, 2000*). An aliquot (10 µL) of an overnight bacterial culture grown in LB (30°C, with shaking at 200 rpm) was added to 2 mL of lactate minimal media with AQDS (5.0 mM) as the sole TEA (no sodium fumarate added). Reduction of AQDS to AHQDS generates a red color. Wild type *S. oneidensis* MR-1 was added to the positive control culture tubes and *menC* mutant was

added to both negative control and test substrate culture tubes. Test substrates – HPLC fractions or pure compounds – were resuspended to the desired concentration (1 mg/mL or 150 µL whichever was higher for HPLC fractions) in 50% ACN/$H_2O$ and then 25 µL was added to the culture tube of a test substrate. The solvent was added to both the positive and negative controls. Upon addition of test substrates and solvent blanks the culture tubes were incubated at room temperature for 24 hr in an anaerobic chamber (Coy Laboratory), at which point they were visually inspected for color changes comparing the culture tubes of the test substrates to both the positive and negative controls.

To quantify the reduction of AQDS, the above method was modified to be compatible with a 96-well format in order to read the production of AHQDS with a plate reader. Each test substrate was resuspended to the desired concentration in 50% ACN/$H_2O$ and 1 µL was added to individual wells in triplicate along with solvent blanks in both the positive and negative control wells. An aliquot (100 µL) of aerobic cultures of both wild type MR-1 and *menC* mutant that were grown overnight in LB at 30°C with 200 rpm shaking, was used to inoculate 20 mL of minimal media that was supplemented with AQDS (5.0 mM) as the sole TEA. This mixture was vortexed and then 199 µL was added to each well – wild type MR-1-containing media to positive control wells and *menC* mutant-containing media to both negative control and test substrate wells. The plates were then incubated in an anaerobic chamber at room temperature for 24 hr before being analyzed on a Molecular Devices M5 plate reader by monitoring absorbance at 408 nm.

## Calculation of ACNQ production and diffusion in an agarose gel

To estimate the local concentration of ACNQ produced by a streak of *Shewanella oneidensis* MR-1 in a 1.5% agarose gel, we assumed that only two processes occurred: production of ACNQ by the bacteria and diffusion of ACNQ through the agarose.

We estimated the production of ACNQ and its diffusion using our data and other data from the literature. To estimate the production rate of ACNQ per cell of *Shewanella oneidensis* (p), we used the fact that the concentration of ACNQ was 1.5 nM after 4 days of production by a stationary phase culture with a cell density of ~$1 \times 10^9$ cell $mL^{-1}$. This calculation yielded p=$4 \times 10^{-23}$ moles $cell^{-1}$ $s^{-1}$. The diffusion coefficient of ACNQ in water has been measured to be $5.2 \times 10^{-6}$ $cm^2$ $s^{-1}$ (*Batchelor-McAuley et al., 2010*). Molecules of similar size and hydrophobicities have a diffusion coefficient in 1.5% agarose that is 95% that of their diffusion coefficient in water (*Fatin-Rouge et al., 2004*), largely because the porosity of agarose is large relative to the size of these molecules. Therefore, we estimate the diffusion coefficient of ACNQ in 1.5% agarose to be D = $5 \times 10^{-6}$ $cm^2$ $s^{-1}$.

Coupled diffusion and production of a molecule is described by the Kolmogorov-Petrovsky-Piskunov equation (*Equation 1*), where C is the concentration of ACNQ (in moles $L^{-1}$), D is the diffusion coefficient of ACNQ (in $cm^2$ $s^{-1}$), and P is the production rate of ACNQ (in moles $s^{-1}$).

$$\frac{dC(x,t)}{dt} = -D\frac{d^2C(x,t)}{dx^2} + P \tag{1}$$

We numerically solved *Equation 1* using an explicit time-domain finite difference formulation (2nd order in space and time) with far-field Dirchelet boundary conditions (concentration of 0) over a large model domain (10 cm) to minimize boundary interaction. Initial conditions were set to C = 0 over the modeling domain and the source term (P) was implemented as an addition to C over each timestep. The formulation was implemented in MATLAB (Mathworks, Cambridge, MA).

## glmS knockout in S. oneidensis MR-1

*glmS* knockout mutation in *S. oneidensis* MR-1 background was made as described previously (*Saltikov and Newman, 2003*). Upstream flanking region of *glmS* was PCR-amplified with GlmS1_fwd (gagcgcgcgtaatacgactcactataggCGTGGCACTTGAAGCTAAG) and GlmS1_rev (attggctttgattACGATTCCGCACATAGTTTTTAC) primers, downstream flanking region was PCR-amplified with GlmS2_fwd (tgtgcggaatcgtAATCAAAGCCAATAAAAAACC) and GlmS2_rev (aaccct-cactaaagggaacaaaagcCGCTGAAGAAGGTAAAGC) primers. pSMV10 (*Saltikov and Newman, 2003*) was PCR amplified with pSMV10_fwd (GCTTTTGTTCCCTTTAGTG) and pSMV10_rev (CCTATAG TGAGTCGTATTACGC). The flanking regions were introduced into pSMV10 using NEBuilder HiFi DNA Assembly Cloning Kit and transformed into chemically competent λ*pir E. coli* UQ950

(*Saltikov and Newman, 2003*). Transformants were selected on 50 µg/mL kanamycin. The construct was isolated and transformed into diaminopimelic acid (DAP) auxotroph *E. coli* donor strain WM3064 (*Saltikov and Newman, 2003*) and plated on LB supplemented with kanamycin and 0.3 mM DAP. For conjugation WM3064 containing the knockout, the construct was grown overnight in LB broth supplemented with kanamycin and DAP. An aliquot (250 µL) of the overnight were washed with 500 µL of LB (Lennox) and mixed with 250 µL of *S. oneidensis* MR-1 overnight. The suspension was plated on LB supplemented with DAP and incubated at 30°C for 8 hr. A streak of the lawn was then spread on LB supplemented with kanamycin, but without DAP. A colony of the trans-conjugant was grown overnight at 30°C in LB broth. *glmS* encodes a putative glutamine-fructose-6-phosphate aminotransferase required for *N*-acetylglucosamine (GlcNAc) biosynthesis, therefore we predicted that the mutant may be a GlcNAc auxotroph. The overnight was diluted 3000 times and 100 µL was spread on LB agar (Lennox) containing 10% (w/v) sucrose with or without 10 mM GlcNAc and grown overnight at 30°C. The next day the plate supplemented with GlcNAc contained approximately twice the number of colonies found on plates without GlcNAc. Colonies from GlcNAc plate were patched on LB, LB supplemented with GlcNAc, or LB supplemented with GlcNAc and kanamycin. Approximately half of the isolates did not proliferate in the absence of GlcNAc. Of these, all were sensitive to kanamycin and contained mutant *glmS* as confirmed by PCR.

## Production and absolute Quantification of ACNQ Production

In order to check for production of ACNQ by different microbial strains - *S. oneidensis* MR-1, its isogenic *menC, menA,* and *glmS* mutants, *V. cholerae* V52, *E. coli* K12, its isogenic *menB, menA,* and *menC* knockout mutants, *Bacteroides fragilis* (DSM 20481), *Lactococcus lactis lactis* (ATCC 25285) - each was grown in 1 L cultures. For absolute quantification, 100 µL of overnight LB cultures of *S. oneidensis* MR-1, isogenic *menC*, and *menA* mutants, *E. coli*, and *V. cholerae* were each used to inoculate 500 mL of lactate minimal media. Cultures were grown under either anaerobic (*B. fragilis* and *L. lactis lactis*) or aerobic (*S. oneidensis*, *V. cholerae*, *E. coli*) conditions for 96 hr. The spent supernatant was then isolated by centrifugation (14,000 rcf, 30 min), filter-sterilized (0.2 µm, Thermo Scientific, cat #: 595–4520), and acidified to a pH of 6.5 with 12.1 M HCl (*aq*). The cell pellets were freeze-dried and weighed to quantify total biomass of each culture. The spent supernatant was then passed through a glass column containing 1 g of Strata-X-A resin (Phenomenex, 33 µm, 85 Å) that was prepped by washing with 15 mL of 100% MeOH and subsequently 15 mL of 100% $H_2O$. Upon passing all of the spent supernatant through the column, the resin was washed with 12 mL of both 25 mM ammonium acetate (*aq*) pH 6.5% and 100% methanol. The dried crude extract was resuspended in 200 µL of 50% ACN/$H_2O$ and 5 µL was quantified using the same HR-LCMS method described above. Integrated extracted ion chromatograms (EIC) for two ion adducts of ACNQ - $[M + H]^+$ and $[M-H_2O + H]^+$ - were summed and compared to a six-point standard curve of synthetically-derived ACNQ in order to obtain the absolute production quantification. Recovery percentages of ACNQ from supernatant to crude extract were also quantified by spiking in 10 nM of $^{15}N$-ACNQ into the spent supernatant of MR-1 cultures prior to the centrifuge step.

## Absolute Quantification of Menaquinone Production

Cultures (100 mL) of bacteria were grown with biological replicates under aerobic conditions in lactate minimal media. Each culture was inoculated with 200 µL of an overnight culture grown in LB and incubated at 30°C, 200 rpm for 72 hr. After which each culture was centrifuged (3900 rcf, 10 min), the supernatant decanted, and the cell pellet was placed on the lyophilizer for 24 hr. The dried cell pellet was transferred and weighed in a 40 mL glass vial, ground, and then extracted with 4.5 mL of 2:1 dichloromethane (DCM)/MeOH for 2 hr while stirring with a magnetic stir bar. The organic solvent was filtered using a glass plug containing celite and dried under vacuum. The crude material was then resuspended in 300 µL of 2:1 isopropanol (IPA)/MeOH and 5 µL of this mixture was analyzed on the HRLCMS where the menaquinone analogs were quantified using the Phenomenex Luna 5 µm C5 100 Å (50 × 4.6 mm) under the following method: hold 100% solvent A for 5 min then quickly gradient to 80% solvent A/20% solvent B over 0.1 min, then gradient to 100% solvent B over 34.9 min with a flow rate of 0.4 mL/min (solvent A: 95% $H_2O$/5% MeOH +0.1% FA with 5 mM ammonium acetate, solvent B: 60% IPA/35% MeOH/5% $H_2O$ + 0.1% FA with 5 mM ammonium acetate). Integrated extracted ion chromatograms for two ion adducts, $[M + H]^+$ and $[M+NH_4]^+$, for each

menaquinone analog were summed and compared to a four-point standard curve of commercially available menaquinone-4 (Sigma Aldrich) in order to obtain the absolute production quantification.

## Relative Quantification of Menaquinone Production under various Culture Conditions

Overnight cultures were grown in LB were centrifuged (3900 rcf, 10 min) and the pellets were rinsed twice with 10 mL DI water. The pellets were then resuspended in 10 mL DI water, where the optical density (OD) was recorded. Larger cultures (3 × 50 mL) were inoculated to a starting OD of 0.05 and grown under the following conditions: 1) aerobically with 30 nM sodium fumarate for 48 hr, 2) anaerobically with 30 nM sodium fumarate for 96 hr, and 3) anaerobically with 20 mM trimethylamine N-oxide (TMAO) for 96 hr in the presence and absence of ACNQ. At either 48 or 96 hr, the OD was measured, and each culture was centrifuged (3900 rcf, 10 min), decanted, rinsed with 10 mL DI water, and re-pelleted. The rinsed pellets were subsequently placed on the lyophilizer for 24 hr and processed using the same protocol described in as the absolute quantification of menaquinone.

## Electrochemical experiments

Dual-chamber electrochemical reactors were used for all the electrochemical experiments. The reactors consisted of a graphite felt working electrode (6.35 mm thick with 16 mm radius, Alfa Aesar) and an Ag/AgCl (3 M KCl, CHI111, CH Instruments) reference electrode in the anode chamber, a Titanium wire (Alfa Aesar) counter electrode in the cathode chamber, chambers were separated by a cation exchange membrane (CMI-7000, Membranes International, Ringwood, NJ). To maintain anaerobic conditions, bioreactors were continuously purged with $N_2$ gas. All the experiment were tested under 30°C. Electrolyte contained M9 growth medium (BD), 50 mM D,L-lactate. Bacteria were cultured aerobically in LB broth at 30°C with 200 rpm shaking overnight. Before injection into the bioreactor, the cells were washed twice with M9 media to remove the LB media.

Chronoamperometry, cyclic voltammetry, and differential pulse voltammetry were carried out using a Bio-Logic Science Instruments potentiostat model VSP-300. For the chronoamperometry test, the applied potential was set at −0.1 V versus Ag/AgCl, electric currents are reported as a function of the geometric surface area of the electrode. Cyclic voltammetry measurements in the potential region of −0.8 to +0.6 V versus Ag/AgCl and a scan rate of 10 mV $s^{-1}$ (if there is no additional declaration). Differential pulse voltammetry measurement was performed with the same scan window, 50 mV pulses height, 500 ms pulses width, 1 mV step height and 1000 ms step time.

## Acknowledgements

This work was funded by NIH [R01 GM086258 and AT980074 (JC)] and work performed at the Molecular Foundry was supported by the Office of Science, Office of Basic Energy Sciences, of the U.S. Department of Energy under Contract No. DE-AC02-05CH11231 (CMAF). LS acknowledges support by the China Scholarship Council (No. 201606090098), CMAF and LS were supported by the Office of Naval Research (N0001419IP00023). EMH was funded by the Ruth L Kirschstein National Research Service Award (5F32GM103010). We thank Buz Barstow at Princeton University for providing *S. oneidensis menA* mutant (*menA*::Himar), The Coli Genetic Stock Center at Yale University for providing all of the *E. coli* mutants (DBI-0742708), and Jeff Granick at the University of Minnesota for providing *S. oneidensis bfe* and *mtr* mutants (Δ*bfe* and Δ*mtr*). We thank J Ajo-Franklin (LBNL) for providing his diffusion modeling implementation. We also thank the ICCB-L core facility in the Biological Chemistry and Molecular Pharmacology department within Harvard Medical School for analytical support.

## Additional information

### Competing interests

Jon Clardy: Reviewing editor, *eLife*. The other authors declare that no competing interests exist.

## Funding

| Funder | Grant reference number | Author |
| --- | --- | --- |
| National Institute of General Medical Sciences | GM086258 | Jon Clardy |
| National Center for Complementary and Integrative Health | AT980074 | Jon Clardy |
| U.S. Department of Energy | DE-AC02-05CH11231 | Caroline M Ajo-Franklin |
| National Institute of General Medical Sciences | 5F32GM103010 | Elissa Hobert |
| China Scholarship Council | 201606090098 | Lin Su |

The funders had no role in study design, data collection and interpretation, or the decision to submit the work for publication.

## Author contributions

Emily Mevers, Conceptualization, Data curation, Formal analysis, Writing—original draft, Writing—review and editing; Lin Su, Data curation, Formal analysis, Writing—original draft, Writing—review and editing; Gleb Pishchany, Moshe Baruch, Jose Cornejo, Elissa Hobert, Eric Dimise, Data curation, Formal analysis, Writing—review and editing; Caroline M Ajo-Franklin, Conceptualization, Supervision, Funding acquisition, Investigation, Writing—original draft, Writing—review and editing; Jon Clardy, Conceptualization, Supervision, Funding acquisition, Writing—original draft, Writing—review and editing

## Author ORCIDs

Emily Mevers https://orcid.org/0000-0001-7986-5610
Lin Su https://orcid.org/0000-0001-8784-3120
Jon Clardy https://orcid.org/0000-0003-0213-8356

## Decision letter and Author response

Decision letter https://doi.org/10.7554/eLife.48054.033
Author response https://doi.org/10.7554/eLife.48054.034

# Additional files

## Supplementary files

• Supplementary file 1. (**A**) Acquisition of non-*E. coli* CGCS bacterial strains. (**B**) *E. coli* strains purchased from Coli Genetic Stock Center.
DOI: https://doi.org/10.7554/eLife.48054.029

• Supplementary file 2. Statistical report for *Figure 3C*.
DOI: https://doi.org/10.7554/eLife.48054.030

• Transparent reporting form
DOI: https://doi.org/10.7554/eLife.48054.031

## Data availability

All data generated or analyzed during this study are included in the manuscript and supporting files.

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
