## [Decision Letter]

[Editors’ note: a previous version of this study was rejected after peer review, but the authors submitted for reconsideration. The first decision letter after peer review is shown below.]

Thank you for submitting your work entitled "An elusive electron shuttle from a facultative anaerobe" for consideration by *eLife*. Your article has been reviewed by three peer reviewers, and the evaluation has been overseen by a Reviewing Editor and a Senior Editor. The following individual involved in review of your submission has agreed to reveal their identity: Jules Butt (Reviewer #4).

Though there is much to appreciate about the insights gained into ACNQ production on a technical level, the study falls short of demonstrating an unambiguous role for ACNQ as an electron shuttle. Without that critical piece, despite the impressiveness of the authors' chemistry, the story is not a good fit for *eLife*. Constructive reviews from three experts are appended below that we hope will help the authors design additional experiments to better test whether ACNQ is an electron shuttle. Should that prove to be the case, we would welcome a new version to be resubmitted to *eLife*. Assuming you are prepared to do this, we will endeavor to assign the same editor and reviewers to consider the new version of your work.

Reviewer #1:

Mevers et al. identify a byproduct of menaquinone biosynthesis (ACNQ) and propose that it functions biologically as an electron shuttle for extracellular electron transfer in *Shewanella*. The authors seem to lack familiarity with literature on *Shewanella* physiology and extracellular electron transfer. Specific growth data is absent in this manuscript, focusing rather on use of AQDS as a proxy for respiration activity, which is problematic for reasons described below. It has previously been established MR-1 is able to cross-feed menaquinone defective mutant strains, which the authors do not discuss and conflicts with data presented here. The evidence that ACNQ plays a specific role in extracellular electron transfer in *Shewanella* is unconvincing. Finally, the authors choose not to present any background on what is known about flavins and their established role as extracellular electron shuttles. This seems odd, considering the wealth of data from multiple labs supporting this role for flavins in *Shewanella* and the focus of this manuscript.

Introduction: The authors misrepresent the 2004 report by Myers and Myers as evidence that a quinone shuttle could not be identified. Rather, this work presented conclusive data that cross-feeding was occurring between MR-1 and a strain defective in menaquinone production (a menB point mutation, in this case). It should be noted that Newman and Kolter (2000) also suggested that their observations could be explained by cross-feeding, but favored a model of electron shuttling. The authors go on to claim that ACNQ addition did not restore menaquinone biosynthesis, however some clarifications in the protocol would be helpful to understand better how the cells were grown and if different concentrations of ACNQ were tried.

Results and Discussion, first paragraph: AQDS is perfectly soluble. Its use to understand extracellular electron transfer is confounded by its ability to cross membranes. For example, a screen designed to identify mutants unable to reduce AQDS identified insertions in TolC-dependent efflux pathways (Shyu et al., 2002, JBact). In the subsection “Cellular Respiration Assay with Anthraquinone-2,6-disulfonate (AQDS)” the authors claim that reduction of AQDS correlates with anaerobic respiration is misleading considering reduction can be facilitated by a wide range of electron transfer reactions, including inside the cell.

Results and Discussion, first paragraph: The authors demonstrate in Figure 1C that ACNQ is able to restore AQDS reduction by a menaquinone mutant. Here the authors call this 'respiration.' Please be specific. I would also encourage the authors to avoid the nomenclature 'MR-1 menC' for the sake of clarity. MR-1 refers to the wild-type strain of *S. oneidensis*. Because menC has been altered, the strain should not technically be referred to as MR-1. The authors also use the ‘*ΔmenC*' nomenclature. This nomenclature is typically reserved for deletion mutants. The menC and menA mutants used here are transposon insertions, not deletions (the authors may need to check with Prof. Barstow on the menA mutant).

Figure 2A: How were ACNQ levels standardized to total number of cells? Were cells grown under identical conditions? In the Materials and methods subsection “Production and Absolute Quantification of ACNQ Production” it would seem that culture density/cell number was not standardized. Without this correction I don't believe the authors can claim one cell produces more ACNQ than another.

Results and Discussion, sixth paragraph: I'm confused by the logic here. The authors wish to explore the potential ability of ACNQ to function as an extracellular electron shuttle. However, the authors then go on to cite an example of where ACNQ clearly has an intracellular role as a mediator to reduce NADP, which is found inside cells.

Results and Discussion, sixth paragraph: Results from the experiment described here are not presented in Figure 1. It is unclear what is actually being tested? Growth? Reduction of AQDS? Are the authors actually growing the cells with AQDS as the terminal electron acceptor or simply monitoring AQDS color change?

Results and Discussion, sixth paragraph: It is unclear how the cells are being grown to test if the menaquinone pool is being replenished by ACNQ. It seems like it may have been aerobic growth? It should be noted that replenishment is precisely what was previously observed by Myers and Myers (2004, AEM). The experiment can be done where cells are grown anaerobically with TMAO or oxygen – electron acceptors that are dependent on ubiquinone and not menaquinone in *Shewanella*. If the authors are using AQDS/fumarate/nitrate/DMSO/FeCitrate as their electron acceptor, the cells may lack the energy needed to convert sufficient ACNQ into MK in order to take advantage of any of these electron acceptors.

Results and Discussion, seventh paragraph: If ACNQ is able to recover a menC mutant's inability to produce menaquinone under the conditions used, I would also expect to see an enhanced rate of current generation. There appears to be a typo in the Y-axis, should this be just 'mA'? Because MR-1 control is not included here it is impossible to know to what extent current generation was restored. Please also describe what the error represents in the measurements and include the ACNQ concentration in the figure legend. The authors should consider including the *Shewanella* electrochemical data alongside (and on the same scale) as their *E. coli* data. This would help illustrate why the addition of a shuttle compound could dramatically enhance activity from the engineered *E. coli* strain.

Results and Discussion, seventh paragraph: One of the biggest problems I have with the work presented here. It seems that sometimes ACNQ addition increased electrochemical activity of MR-1 cultures and sometimes it did not. The authors blame this variability on *Shewanella's* ability to breathe many different electron acceptors. It seems more likely that the authors were unable to see the effect they were looking for. It is therefore difficult to make an augment that ACNQ plays any direct role in extracellular electron transfer.

Figure 3: The enhancement of the engineered *E. coli* strain is interesting, but it is still unclear what exactly is being observed here. Please include 'CymA' in the figure legend here and a description of what the error bars are. CymA is important because it is known to require a menaquinone-derived cofactor for activity (McMillan et al., 2012, JBC). The observed enhancement here could be related to helping provide a cofactor that may be limiting in the native *E. coli* system? Did the authors attempt to quantify quinone levels after ACNQ addition? What is the current profile when ACNQ is added without cells? Is oxygen being scrubbed from the system and therefore more electrons are available to the electrode? Importantly, the authors should perform a medium swap experiment. If the enhancement was due to a co-factor addition and/or quinone pool augmentation current generation will remain high after the swap. If ACNQ is indeed a shuttle, current will return to the unaugmented levels. Finally, considering the established role for riboflavin and FMN as extracellular electron shuttles, it would be interesting to see how addition of these compounds assist in current production under these conditions.

Reviewer #2:

This contribution from Clardy and co-workers addresses a longstanding mystery in natural product isolation: identification of the small molecule that complements respiration in menaquinone-deficient mutants of *Shewanella oneidensis* MR-1. Using a bioactivity-guided approach, they isolate ACNQ and confirm its structure via comparison to a synthetic standard. They investigate the biosynthesis of this molecule, providing compelling evidence that it derived from a non-enzymatic transformation of the menaquinone biosynthetic intermediate DHNA. Finally, they show that addition of ACNQ recovers current generation in the *S. oneidensis* menC mutant and instantaneously increases current generation by an *E. coli* strain expressing CymA and MtrCAB. Together, these results suggest ACNQ is indeed the 'missing' redox-active small molecule originally identified by Newman and Kolter.

My main concern related to the claims of this paper is that the evidence presented for ACNQ playing a direct role in extracellular transfer to insoluble terminal electron acceptors is quite limited. For example, the first paragraph of the Results and Discussion describes AQDS as an insoluble electron acceptor. I am unclear if this is a typo, but AQDS is used as a model compound as a soluble EET mediator. It is extremely soluble in water. It is an excellent choice for the authors' screen to discover ACNQ, but reduction of AQDS does not necessarily demonstrate an extracellular role for ACNQ. Indeed, in the sixth paragraph of the Results and Discussion the authors acknowledge that AQDS can be reduced by intracellular enzymes. Although the authors have shown that ACNQ does not restore the menaquinone pool, I do not think they have eliminated the possibility that ACNQ is complementing the intracellular reduction of AQDS. Similarly, the conclusion states "[ACNQ] alone is capable of recovering respiration on metal-oxides". But the manuscript does not present data to support this conclusion-as far as I can tell, the only metal substrate tested was iron citrate, a soluble form of chelated iron (and I cannot seem to find the data for iron citrate or fumarate experiments).

There is also little discussion of this result in the context of what is already known about flavins, which are currently proposed to be responsible for extracellular electron transport in S. oneidensis (see Marsili, Baron, Shikhare, Coursolle, Gralnick, and Bond 2008 PNAS; Kotloski and Gralnick 2013 mBio). Indeed, flavins are mentioned only in the final sentence of the paper. Can the authors reconcile their results with what has been previously reported? For example, have authors ruled out the possibility that ACNQ complements the ability of S. oneidensis to transfer electrons to secreted flavins?

Overall, I feel this is a very important finding that should be communicated. While evidence to support a role for ACNQ's involvement in extracellular electron transfer may still be lacking, this is certainly a complicated system and I think additional experiments may be beyond the scope of this paper. But I do feel that the authors should revise their Introduction and Results and Discussion sections to better situate their findings in the context of what is known about flavin redox shuttles and clearly explain any limitations of their results.

Reviewer #3:

The manuscript presents a rigorous study of an excreted quinone that is most likely that previously reported to contribute to extracellular electron transfer by *S. putrefaciens* MR-1. A wealth of analytical and biochemical data support the authors' conclusion that this molecule is ACNQ, formed spontaneously by deamination of DHNA which is an intermediate in the pathway of menaquinone synthesis. The research presents a significant advance in our knowledge of contributors to extracellular electron transfer. Furthermore, the results set the scene for studies that will be needed to establish how redox transformation of ACNQ is coupled to intracellular redox events. Here the authors address a potential contribution to biotechnology by considering the contribution of ACNQ to anodic current generation by lactate growing cells. ACNQ recovery of electron exchange between lactate grown ΔmenC MR-1 and an electrode (Supplementary Figure 23) supports of role for ACNQ in that context. Equivalent data for an MtrCAB, CymA containing *E. coli* strain were presented in the main paper (Figure 3). I could not see the benefit of presenting the studies with the *E. coli* strain unless equivalent data for other experiments (i.e., currents +/- ACNQ for host *E. coli* strain) are included with the aim of elucidating whether CymA, MtrCAB are needed for ACNQ current stimulation. Of course these are interesting questions – but perhaps too much to address in this short contribution?

The results for the experiments of growing ΔmenC MR-1 on iron citrate and fumarate (Results and Discussion, sixth paragraph) with ACNQ, DHNA, riboflavin and menadione should be included in the Supporting Information. The results for fumarate reduction are particularly interesting since fumarate is reduced in the periplasm by a dedicated enzyme FccA. To stimulate reduction of that terminal electron acceptor there would seem to be no need for ACNQ to act as an extracellular shuttle.

I was not familiar with the type of assay presented in Figure 1A and would have appreciated a much clearer figure legend to that key data set. For readers' benefit I would urge the authors to revise the figure legends throughout the Supporting Information so they are 'stand alone'.

[Editors’ note: what now follows is the decision letter after the authors submitted for further consideration.]

Thank you for resubmitting your work entitled "An elusive electron shuttle from a facultative anaerobe" for further consideration at *eLife*. Your revised article has been favorably evaluated by Gisela Storz as the Senior Editor, and three reviewers, one of whom is a member of our Board of Reviewing Editors.

The manuscript is substantially improved and the authors are to be commended for being so responsive to the constructive set of first reviews. However, there is one large remaining conceptual sticking point: at natural levels produced by the WT under the conditions of the experiments, ACNQ does not appear to act as an electron shuttle. What then does it mean for electron shuttling to be stimulated at non-physiological levels? Bringing this observation to the foreground would go far to clarify the impact of this finding on the EET field. Clarifying this point should be straightforward with one more round of revision and will substantially improve the contribution this study will make the to literature. Please also consider the constructive points of the reviewers below before submitting a final revision.

Reviewer #1:

To my reading, the revised manuscript presents a coherent narrative that addresses my concerns of the original submission. Identification and biosynthesis of ACNQ is described followed by demonstration that ACNQ can enhance electron exchange between MR-1 and electrodes. Importantly the electrochemical results are more substantive and convincing in the role of ACNQ as a shuttle between electrode and bacteria. Where content in the original manuscript was less convincing it has been removed.

Reviewer #2:

The manuscript is improved. The authors have put together an impressive array of experiments. I believe they have found something interesting in ACNQ – a byproduct of menaquinone biosynthesis that appears to be able to 'steal' electrons from cells. However, the claim that this molecule plays a significant role in extracellular electron transfer in *Shewanella oneidensis* is not supported by the results presented here.

When the authors poise a graphite felt electrode at -0.1V vs Ag/AgCl and add 1 μm ACNQ (~ 1000X more than is produced natively) a small amount of current is produced. The new chronoamperometry data is nice, showing that ACNQ indeed appears to act as an electron shuttle – current drops when it's taken away and is restored when added back. However, this current is independent of the Mtr pathway or menaquinone biosynthesis, both of which are essential for growth of *S. oneidensis* via extracellular electron transfer on a variety of substrates (electrodes, iron and manganese minerals, Fe-citrate, etc.). When ACNQ is produced at wild-type levels, MR-1 is unable to produce current under these conditions (-0.1V vs. Ag/AgCl) as the authors point out (Results and Discussion, ninth paragraph). The authors demonstrate that this molecule can act as an electron shuttle, but also show that this activity is irrelevant at the levels naturally produced by the cells. In my opinion, the authors should do a better job communicating this observation.

ACNQ appears to be acting as an electron pirate, perhaps in a way analogous to neutral red, providing a sink for cellular electrons acquired in the periplasm, cytoplasmic membrane or in the cytoplasm, then delivering the electrons to the poised graphite felt electrode (Park and Zeikus, 2000, AEM). Like neutral red, this activity could have some interesting biotechnological applications.

A previous comment of mine and response is pasted below. If ACNQ is known to enter cells and acquire electrons from molecules such as NADPH, perhaps this explains the men and mtr-independent current generation? This possibility seems relevant to understanding the role of ACNQ in electron transfer and could be similar to what has been observed previously with neutral red.

Results and Discussion, sixth paragraph: I'm confused by the logic here. The authors wish to explore the potential ability of ACNQ to function as an extracellular electron shuttle. However, the authors then go on to cite an example of where ACNQ clearly has an intracellular role as a mediator to reduce NADP, which is found inside cells.

---

## [Author Response]

[Editors’ note: the author responses to the first round of peer review follow.]

Reviewer #1:Mevers et al. identify a byproduct of menaquinone biosynthesis (ACNQ) and propose that it functions biologically as an electron shuttle for extracellular electron transfer in Shewanella. The authors seem to lack familiarity with literature on Shewanella physiology and extracellular electron transfer. Specific growth data is absent in this manuscript, focusing rather on use of AQDS as a proxy for respiration activity, which is problematic for reasons described below. It has previously been established MR-1 is able to cross-feed menaquinone defective mutant strains, which the authors do not discuss and conflicts with data presented here. The evidence that ACNQ plays a specific role in extracellular electron transfer in Shewanella is unconvincing. Finally, the authors choose not to present any background on what is known about flavins and their established role as extracellular electron shuttles. This seems odd, considering the wealth of data from multiple labs supporting this role for flavins in Shewanella and the focus of this manuscript.Introduction: The authors misrepresent the 2004 report by Myers and Myers as evidence that a quinone shuttle could not be identified. Rather, this work presented conclusive data that cross-feeding was occurring between MR-1 and a strain defective in menaquinone production (a menB point mutation, in this case). It should be noted that Newman and Kolter, 2000, also suggested that their observations could be explained by cross-feeding, but favored a model of electron shuttling. The authors go on to claim that ACNQ addition did not restore menaquinone biosynthesis, however some clarifications in the protocol would be helpful to understand better how the cells were grown and if different concentrations of ACNQ were tried.

We thank the reviewer for making this distinction. We have clarified our language to more accurately represent the discoveries in the 2004 Myers and Myers paper. In addition, we have expanded our description of how we probed whether ACNQ could result in MQ production.

Results and Discussion, first paragraph: AQDS is perfectly soluble. Its use to understand extracellular electron transfer is confounded by its ability to cross membranes. For example, a screen designed to identify mutants unable to reduce AQDS identified insertions in TolC-dependent efflux pathways (Shyu et al., 2002, JBact). In the subsection “Cellular Respiration Assay with Anthraquinone-2,6-disulfonate (AQDS)” the authors claim that reduction of AQDS correlates with anaerobic respiration is misleading considering reduction can be facilitated by a wide range of electron transfer reactions, including inside the cell.

We thank the reviewer for these corrections. We have corrected our description of AQDS throughout the manuscript.

Results and Discussion, first paragraph: The authors demonstrate in Figure 1C that ACNQ is able to restore AQDS reduction by a menaquinone mutant. Here the authors call this 'respiration.' Please be specific. I would also encourage the authors to avoid the nomenclature 'MR-1 menC' for the sake of clarity. MR-1 refers to the wild-type strain of S. oneidensis. Because menC has been altered, the strain should not technically be referred to as MR-1. The authors also use the 'ΔmenC' nomenclature. This nomenclature is typically reserved for deletion mutants. The menC and menA mutants used here are transposon insertions, not deletions (the authors may need to check with Prof. Barstow on the menA mutant).

We thank the reviewer for these corrections. We have clarified our language throughout the manuscript to differentiate between observations of growth and reduction of a TEA. We have also corrected our notation, referring to the transposon insertions as *menA*::Himar and *menC*::Tn10.

Figure 2A: How were ACNQ levels standardized to total number of cells? Were cells grown under identical conditions? In the Materials and methods subsection “Production and Absolute Quantification of ACNQ Production” it would seem that culture density/cell number was not standardized. Without this correction I don't believe the authors can claim one cell produces more ACNQ than another.

The reviewer makes a good point here. All cells were grown under identical conditions, though starting culture density was not controlled. This was an oversight, but we did standardize the ACNQ levels to total cell biomass. Additionally, we have clarified the growth conditions of each bacterial strain in the Materials and methods section.

Results and Discussion, sixth paragraph: I'm confused by the logic here. The authors wish to explore the potential ability of ACNQ to function as an extracellular electron shuttle. However, the authors then go on to cite an example of where ACNQ clearly has an intracellular role as a mediator to reduce NADP, which is found inside cells.

We thank the reviewer for sharing their opinion, we have opted to remove these lines from the manuscript as they are not directly relevant.

Results and Discussion, sixth paragraph: Results from the experiment described here are not presented in Figure 1. It is unclear what is actually being tested? Growth? Reduction of AQDS? Are the authors actually growing the cells with AQDS as the terminal electron acceptor or simply monitoring AQDS color change?

We thank the reviewer for pointing out this lack of clarity, and we have edited the manuscript to more clearly identify the measurements. Figure 1C shows changes in AQDS color change.

Results and Discussion, sixth paragraph: It is unclear how the cells are being grown to test if the menaquinone pool is being replenished by ACNQ. It seems like it may have been aerobic growth? It should be noted that replenishment is precisely what was previously observed by Myers and Myers (2004, AEM). The experiment can be done where cells are grown anaerobically with TMAO or oxygen – electron acceptors that are dependent on ubiquinone and not menaquinone in Shewanella. If the authors are using AQDS/fumarate/nitrate/DMSO/FeCitrate as their electron acceptor, the cells may lack the energy needed to convert sufficient ACNQ into MK in order to take advantage of any of these electron acceptors.

Myers and Myers saw that addition of either MK-4 or DHNA were able to recover Fe(III) reduction in a menaquinone-negative mutant. In line with their observations, we observe addition of DHNA to the menaquinone-negative mutant recovers AQDS reduction (Figure 1C). We also observe that addition of ACNQ to the *S. oneidensis menC::Tn10*, which cannot produce either MQ or ACNQ, recovers AQDS reduction (Figure 1C). As the reviewer points out, this recovery could arise for three reasons: (i) ACNQ stimulates production of MK, which is necessary for AQDS reduction, and ACNQ is not necessary for AQDS reduction, (ii) ACNQ acts as a redox shuttle, which is necessary for AQDS reduction and MK is not necessary for AQDS reduction, or (iii) ACNQ stimulates MQ production and both MK and ACNQ are needed for AQDS reduction.

Our data, including new experiments added to the resubmission, is most consistent with hypothesis (ii) that ACNQ acts as redox shuttle which is necessary for AQDS reduction and that MK is not necessary for this AQDS reduction. Firstly, when we grow *S. oneidensis menC::Tn10* supplemented with 0.05-1 μM of ACNQ under anaerobic conditions with either TMAO or fumarate as the sole terminal electron acceptors, we see no regeneration of MK. Secondly, no significant changes in the MK pools were observed for any of the bioreactor cultures, i.e. MR-1 biofilms that were treated with ACNQ had comparably levels to MR-1 untreated and virtually no MK was detected in *S. oneidensis menC::Tn10 c*ultures regardless if they were treated with ACNQ or not.

Results and Discussion, seventh paragraph: If ACNQ is able to recover a menC mutant's inability to produce menaquinone under the conditions used, I would also expect to see an enhanced rate of current generation. There appears to be a typo in the Y-axis, should this be just 'mA'? Because MR-1 control is not included here it is impossible to know to what extent current generation was restored. Please also describe what the error represents in the measurements and include the ACNQ concentration in the figure legend. The authors should consider including the Shewanella electrochemical data alongside (and on the same scale) as their E. coli data. This would help illustrate why the addition of a shuttle compound could dramatically enhance activity from the engineered E. coli strain.

We have removed all electrochemical data for the engineered *E. coli* strains and replaced it with experiments using *S. oneidensis* MR-1 and *S. oneidensis menC::Tn10* (see Figure 3). These new data more strongly supports our conclusion that ACNQ is acting as an electron shuttle. We have now shown that addition of ACNQ alone increases current generation in *S. oneidensis menC::Tn10* independent of menaquinone and cyt c.

Results and Discussion, seventh paragraph: One of the biggest problems I have with the work presented here. It seems that sometimes ACNQ addition increased electrochemical activity of MR-1 cultures and sometimes it did not. The authors blame this variability on Shewanella's ability to breathe many different electron acceptors. It seems more likely that the authors were unable to see the effect they were looking for. It is therefore difficult to make an augment that ACNQ plays any direct role in extracellular electron transfer.

Our poor choice of words misled the reviewer on this point. We consistently observed that ACNQ significantly increases current production of the *menC::Tn10* strain relative to the background current. However, the magnitude of the current production in our first experiments was relatively variable, as indicated by the large error bars shown in our old Supplementary Figure 23. We have successfully overcome this issue and our revised manuscript only includes experiments conducted using *S. oneidensis* MR-1 and *S. oneidensis menC::Tn10*.

Figure 3: The enhancement of the engineered E. coli strain is interesting, but it is still unclear what exactly is being observed here. Please include 'CymA' in the figure legend here and a description of what the error bars are. CymA is important because it is known to require a menaquinone-derived cofactor for activity (McMillan et al., 2012, JBC). The observed enhancement here could be related to helping provide a cofactor that may be limiting in the native E. coli system? Did the authors attempt to quantify quinone levels after ACNQ addition? What is the current profile when ACNQ is added without cells? Is oxygen being scrubbed from the system and therefore more electrons are available to the electrode? Importantly, the authors should perform a medium swap experiment. If the enhancement was due to a co-factor addition and/or quinone pool augmentation current generation will remain high after the swap. If ACNQ is indeed a shuttle, current will return to the unaugmented levels. Finally, considering the established role for riboflavin and FMN as extracellular electron shuttles, it would be interesting to see how addition of these compounds assist in current production under these conditions.

The reviewer points out that the current increase observed could be due to shuttling of electrons from abiotic electron donors in the media between ACNQ and the electrode or could be due to ACNQ-triggered production of MK inside *E. coli*. We have performed new experiments (new Figure 3) to show that abiotic addition of ACNQ does not reduce the electrode. We have also measured the MK content and show that it is unaffected by addition of ACNQ. These experiments thus exclude these alternative causes for the observed current increase in *S. oneidensis* MR-1 or *S. oneidensis menC::Tn10.*

In addition, per the reviewer’s suggestion, we have now performed a media swap experiment with the *S. oneidensis* MR-1. The results of this media swap is outlined in Figure 3A, and they strongly indicate that the original oxidation current catalyzed by MR-1 arises from an ACNQ-related reaction.

Reviewer #2:[…] My main concern related to the claims of this paper is that the evidence presented for ACNQ playing a direct role in extracellular transfer to insoluble terminal electron acceptors is quite limited. For example, the first paragraph of the Results and Discussion describes AQDS as an insoluble electron acceptor. I am unclear if this is a typo, but AQDS is used as a model compound as a soluble EET mediator. It is extremely soluble in water. It is an excellent choice for the authors' screen to discover ACNQ, but reduction of AQDS does not necessarily demonstrate an extracellular role for ACNQ. Indeed, in the sixth paragraph of the Results and Discussion the authors acknowledge that AQDS can be reduced by intracellular enzymes. Although the authors have shown that ACNQ does not restore the menaquinone pool, I do not think they have eliminated the possibility that ACNQ is complementing the intracellular reduction of AQDS. Similarly, the conclusion states "[ACNQ] alone is capable of recovering respiration on metal-oxides". But the manuscript does not present data to support this conclusion-as far as I can tell, the only metal substrate tested was iron citrate, a soluble form of chelated iron (and I cannot seem to find the data for iron citrate or fumarate experiments).

As the reviewer points out AQDS is indeed a soluble terminal electron acceptor, and the conclusion indicated in the seventh paragraph of the Results and Discussion does not derive from our data using AQDS. Instead the data shown in Figure 3 implicates ACNQ in reduction of an electrode.

There is also little discussion of this result in the context of what is already known about flavins, which are currently proposed to be responsible for extracellular electron transport in S. oneidensis (see Marsili, Baron, Shikhare, Coursolle, Gralnick, and Bond 2008 PNAS; Kotloski and Gralnick, 2013). Indeed, flavins are mentioned only in the final sentence of the paper. Can the authors reconcile their results with what has been previously reported? For example, have authors ruled out the possibility that ACNQ complements the ability of S. oneidensis to transfer electrons to secreted flavins?

*cymAmtrCAB E. coli* doesn’t secrete flavins so that cannot be the origin of the increase in current shown in the old Figure 3, but removed in the revised manuscript. We have conducted additional experiments that show that current generation in *S. oneidensis* has a result to supplemented ACNQ is primarily a result of extracellular interactions.

Overall, I feel this is a very important finding that should be communicated. While evidence to support a role for ACNQ's involvement in extracellular electron transfer may still be lacking, this is certainly a complicated system and I think additional experiments may be beyond the scope of this paper. But I do feel that the authors should revise their Introduction and Results and Discussion sections to better situate their findings in the context of what is known about flavin redox shuttles and clearly explain any limitations of their results.

We have expanded the Introduction and Results and Discussion sections to hopefully improve the context of this new work.

Reviewer #3:The manuscript presents a rigorous study of an excreted quinone that is most likely that previously reported to contribute to extracellular electron transfer by S. putrefaciens MR-1. A wealth of analytical and biochemical data support the authors' conclusion that this molecule is ACNQ, formed spontaneously by deamination of DHNA which is an intermediate in the pathway of menaquinone synthesis. The research presents a significant advance in our knowledge of contributors to extracellular electron transfer. Furthermore, the results set the scene for studies that will be needed to establish how redox transformation of ACNQ is coupled to intracellular redox events. Here the authors address a potential contribution to biotechnology by considering the contribution of ACNQ to anodic current generation by lactate growing cells. ACNQ recovery of electron exchange between lactate grown ΔmenC MR-1 and an electrode (Supplementary Figure 23) supports of role for ACNQ in that context. Equivalent data for an MtrCAB, CymA containing E. coli strain were presented in the main paper (Figure 3). I could not see the benefit of presenting the studies with the E. coli strain unless equivalent data for other experiments (i.e., currents +/- ACNQ for host E. coli strain) are included with the aim of elucidating whether CymA, MtrCAB are needed for ACNQ current stimulation. Of course these are interesting questions – but perhaps too much to address in this short contribution?

We thank the reviewer for their thoughtful comments. We have removed all electrochemical experiments that used the modified *E. coli* strains and conducted similar experiments using *S. oneidensis* MR-1 and *S. oneidensis menC::Tn10*. Thus far, we have been unable to elucidate exactly how ACNQ is functioning, i.e. through CymA or MtrCAB, and further studies will be needed to elucidate ACNQ’s exact role.

The results for the experiments of growing ΔmenC MR-1 on iron citrate and fumarate (Results and Discussion, sixth paragraph) with ACNQ, DHNA, riboflavin and menadione should be included in the Supporting Information. The results for fumarate reduction are particularly interesting since fumarate is reduced in the periplasm by a dedicated enzyme FccA. To stimulate reduction of that terminal electron acceptor there would seem to be no need for ACNQ to act as an extracellular shuttle.

This line has been removed from the revised manuscript.

I was not familiar with the type of assay presented in Figure 1A and would have appreciated a much clearer figure legend to that key data set. For readers' benefit I would urge the authors to revise the figure legends throughout the Supporting Information so they are 'stand alone'.

Additional descriptions have been added to this figure legend and those throughout the SI to help improve clarity.

[Editors' note: the author responses to the re-review follow.]

Thank you for resubmitting your work entitled "An elusive electron shuttle from a facultative anaerobe" for further consideration at eLife. Your revised article has been favorably evaluated by Gisela Storz as the Senior Editor, and three reviewers, one of whom is a member of our Board of Reviewing Editors.The manuscript is substantially improved and the authors are to be commended for being so responsive to the constructive set of first reviews. However, there is one large remaining conceptual sticking point: at natural levels produced by the WT under the conditions of the experiments, ACNQ does not appear to act as an electron shuttle. What then does it mean for electron shuttling to be stimulated at non-physiological levels? Bringing this observation to the foreground would go far to clarify the impact of this finding on the EET field. Clarifying this point should be straightforward with one more round of revision and will substantially improve the contribution this study will make the to literature. Please also consider the constructive points of the reviewers below before submitting a final revision.

Bulk levels of ACNQ in mature liquid cultures (1.5 nM) are only about 12-20x lower than measured EC50 value (25 nM) for reduction of AQDS. Under physiologically relevant growth conditions where *S. oneidensis* MR-1 is growing in a biofilm on a mineral, the local concentration will be significantly higher than the bulk concentration in culture conditions. As part of this revision, we did quantify the local concentration of ACNQ using a simulation (see SI for details) and this simulation suggests that the local concentration of ACNQ when growing in a biofilm would be 1000-fold greater than the bulk concentration. Even if this calculation is off by one or two orders of magnitude, the local concentration, which is the physiologically relevant concentration, would still be higher than the EC50. The question equates bulk and physiological concentrations, which are different.

We used 1 μM ACNQ in the bioreactors in order to ensure that we maintained levels above the EC90 throughout the entire multi-day experiment, i.e. accounting for any potential degradation of ACNQ, and never titrated down the concentration of ACNQ. We have emphasized these important points in the text.

Reviewer #2:The manuscript is improved. The authors have put together an impressive array of experiments. I believe they have found something interesting in ACNQ – a byproduct of menaquinone biosynthesis that appears to be able to 'steal' electrons from cells. However, the claim that this molecule plays a significant role in extracellular electron transfer in Shewanella oneidensis is not supported by the results presented here.When the authors poise a graphite felt electrode at -0.1V vs. Ag/AgCl and add 1 μm ACNQ (~ 1000X more than is produced natively) a small amount of current is produced. The new chronoamperometry data is nice, showing that ACNQ indeed appears to act as an electron shuttle – current drops when it's taken away and is restored when added back. However, this current is independent of the Mtr pathway or menaquinone biosynthesis, both of which are essential for growth of S. oneidensis via extracellular electron transfer on a variety of substrates (electrodes, iron and manganese minerals, Fe-citrate, etc.). When ACNQ is produced at wild-type levels, MR-1 is unable to produce current under these conditions (-0.1V vs. Ag/AgCl) as the authors point out (Results and Discussion, ninth paragraph). The authors demonstrate that this molecule can act as an electron shuttle, but also show that this activity is irrelevant at the levels naturally produced by the cells. In my opinion, the authors should do a better job communicating this observation.

We thank the reviewer for the thoughtful comments; however, we believe they misunderstood our results. Both the wild type and mutant strains were washed several times prior to inoculation into the bioreactors in order to avoid transferring excreted ACNQ or flavins. In addition, we’d like to point out the -0.1 V vs. Ag/AgCl potential applied the electrode is not ideal for MR1 to generate energy through EET anaerobically as we “turn off” the direct EET through Mtr pathway. In addition, there is no mediator available to perform indirect EET after washing. Most likely these factors will repress the regeneration of both ACNQ and flavin below the natural levels. As expected, no current was generated in the absence of any small molecule mediators. We did not test whether or not unwashed wild type strain, with flavin and/or ACNQ present, could support bacteria growth.

We chose to use significantly higher concentrations of ACNQ in the bioreactors in order to ensure that we maintained levels above the EC_90_ throughout the entire multi-day experiment, i.e. accounting for any potential degradation of ACNQ. Furthermore, levels of ACNQ in liquid culture (1.5 nM) is only about 12-20x lower than measured EC_50_ value (25 nM) for reduction of AQDS. Under physiologically relevant growth conditions where *S. oneidensis* MR-1 is growing in a biofilm on a mineral, the local concentration is presumably significantly higher than the bulk concentration.

ACNQ appears to be acting as an electron pirate, perhaps in a way analogous to neutral red, providing a sink for cellular electrons acquired in the periplasm, cytoplasmic membrane or in the cytoplasm, then delivering the electrons to the poised graphite felt electrode (Park and Zeikus, 2000, AEM). Like neutral red, this activity could have some interesting biotechnological applications.A previous comment of mine and response is pasted below. If ACNQ is known to enter cells and acquire electrons from molecules such as NADPH, perhaps this explains the men and mtr-independent current generation? This possibility seems relevant to understanding the role of ACNQ in electron transfer and could be similar to what has been observed previously with neutral red.Results and Discussion, sixth paragraph: I'm confused by the logic here. The authors wish to explore the potential ability of ACNQ to function as an extracellular electron shuttle. However, the authors then go on to cite an example of where ACNQ clearly has an intracellular role as a mediator to reduce NADP, which is found inside cells.

We appreciate the reviewer’s comment, but note that NADP/NADPH has a more negative redox potential (-0.370 V vs. SHE) than ACNQ (-0.32 V vs. Ag/AgCl), which means that at least in principle ACNQ could receive electrons through NADPH directly thus bypassing MK as we established in our studies. We mentioned this and cited studies with cell-free extracts of *B. longum* 6001 in the first submission, which we decided to remove in previous revision based on reviewer comments, and we the fact that we did not attempt to replicate those experiments.

ACNQ has significant promise for biotechnological applications, which is one of the reasons we’d like to share these results with the community.